# Detection of Clinical and Subclinical Lumpy Skin Disease Using Ear Notch Testing and Skin Biopsies

**DOI:** 10.3390/microorganisms9102171

**Published:** 2021-10-19

**Authors:** Laetitia Aerts, Andy Haegeman, Ilse De Leeuw, Wannes Philips, Willem Van Campe, Isabelle Behaeghel, Laurent Mostin, Kris De Clercq

**Affiliations:** 1European Reference Laboratory (EURL) for Diseases Caused by Capripox Viruses, Scientific Directorate Infectious Diseases in Animals, Sciensano, Groeselenberg 99, B-1180 Brussels, Belgium; Laetitia.Aerts@sciensano.be (L.A.); wannes.philips@sciensano.be (W.P.); 2Unit of Exotic and Particular Diseases, Scientific Directorate Infectious Diseases in Animals, Sciensano, Groeselenberg 99, B-1180 Brussels, Belgium; Andy.Haegeman@sciensano.be (A.H.); Ilse.DeLeeuw@sciensano.be (I.D.L.); 3Experimental Center Machelen, Scientific Directorate Infectious Diseases in Animals, Sciensano, Kerklaan 68, B-1830 Machelen, Belgium; Willem.VanCampe@sciensano.be (W.V.C.); Laurent.Mostin@sciensano.be (L.M.); 4Scientific Directorate Infectious Diseases in Animals, Sciensano, Groeselenberg 99, B-1180 Brussels, Belgium; Isabelle.Behaeghel@sciensano.be

**Keywords:** lumpy skin disease, subclinical infection, virus detection, skin biopsies, ear notch testing

## Abstract

Lumpy skin disease (LSD) diagnosis is primarily based on clinical surveillance complemented by PCR of lesion crusts or nodule biopsies. Since LSD can be subclinical, the sensitivity of clinical surveillance could be lower than expected. Furthermore, real-time PCR for the detection of LSD viral DNA in blood samples from subclinical animals is only intermittently positive. Therefore, this study aimed to investigate an acceptable, easily applicable and more sensitive testing method for the detection of clinical and subclinical LSD. An animal experiment was conducted to investigate ear notches and biopsies from unaffected skin taken from the neck and dorsal back as alternatives to blood samples. It was concluded that for early LSD confirmation, normal skin biopsies and ear notches are less fit for purpose, as LSDV DNA is only detectable in these samples several days after it is detectable in blood samples. On the other hand, blood samples are less advisable for the detection of subclinical animals, while ear notches and biopsies were positive for LSD viral DNA in all subclinically infected animals by 16 days post infection. In conclusion, ear notches could be used for surveillance to detect subclinical animals after removing the clinical animals from a herd, to regain trade by substantiating the freedom of disease or to support research on LSDV transmission from subclinical animals.

## 1. Introduction

Lumpy skin disease (LSD) is a viral infectious disease of cattle and buffalo caused by lumpy skin disease virus (LSDV). LSDV is classified into the genus *Capripoxvirus* of the Poxviridae family together with sheeppox virus and goatpox virus. LSD is characterized by multifocal cutaneous nodules and lesions on the mucous membranes of the respiratory and digestive tracts, along with fever, weight loss and depression. LSD is associated with moderate-to-high morbidity but generally low mortality [1]. Still, the disease can have an important socioeconomic impact due to the reduced milk and meat production, inferior hide quality, reduced reproduction due to increased infertility and abortion, reduced draft power of animals and serious trade restrictions in previously free countries [2,3,4]. The severity of the clinical disease is often influenced by the animal’s age, breed, immune status and production period [1]. The main mode of transmission of LSDV is mechanical via blood-feeding insects with frequent feeding habits [5,6], although the main vector is likely to vary between geographical regions and ecosystems [1]. LSD diagnosis is primarily based on the clinical diagnosis of LSD, confirmed by the PCR analysis of lesion crusts or biopsies of the nodules or affected skin [7]. However, LSDV infection is not always apparent, as mild and subclinical disease occur. Even when cattle are experimentally infected, up to 50% of animals remain uninfected or subclinically infected [5,6,8,9,10].

Since its discovery in 1929 in Zambia, LSD became widespread throughout Africa. However, LSD has remained mostly confined to Africa for over 80 years, occasionally leading to outbreaks in the Middle Eastern region. LSD reached Israel in 2012 and Turkey in 2013 [11], and spread to the northern part of Cyprus in 2014 and Greece in 2015 [12], subsequently spreading across the Balkan region in 2016 [13], as well as to the northern Caucasus and the Russian Federation [14,15]. LSD was introduced to the Indian subcontinent and China in 2019, and to other South, East and Southeast Asian countries in 2020–2021 [1,13,14]. This wide dissemination of LSD in the last decade is likely due to a combination of factors, including increased globalization with legal and illegal trading of live animals, migration due to political unrest in several regions and limited access to effective vaccines [1,13,14]. Vaccination is considered the only effective method to control the disease, combined with movement restrictions and the removal of affected animals [16,17]. In addition, awareness campaigns, vector control, increased farm biosecurity and clinical surveillance programs are needed to avoid (re-)introduction. Since LSD can be mild and subclinical [5,6], the sensitivity of clinical surveillance could be lower than expected and must therefore be complemented with laboratory diagnosis [7]. However, farmers are reluctant to have biopsies taken. On the other hand, farmers in the Unites States and the European Union (EU) are used to have ear notches taken in the framework of bovine viral diarrhea (BVD) disease control to detect and eliminate persistently infected calves [18,19]. This kind of sampling and testing might even become more important in the EU as BVD is integrated in the new EU Animal Health Law (Regulation EU 2016/429). The current study aimed at investigating the applicability of ear notch testing for the detection of LSDV in animals with subclinical LSD infection in comparison with tests to confirm the presence of LSDV in clinically diseased animals. In this study, ear notch samples, as well as skin biopsies of unaffected skin taken following the experimental LSDV infection of cattle, were examined using real-time PCR.

## 2. Materials and Methods

### 2.1. Animal Inoculation with LSDV

The LSDV inoculation strain LSD/OA3-Ts.MORAN was kindly provided by the Kimron Veterinary Institute, Israel and the Israeli Veterinary Services, Beit Dagan, Israel, LSD/OA3-Ts.MORAN was cultured on OA3.Ts as described by Babiuk et al. (2007) [20]. The ovine testis cell line OA3.Ts cells (ATCC-CRL-6546) were cultured in DMEM (Thermo Fisher Scientific, Merelbeke, Belgium) supplemented with 10% foetal calf serum (FCS; Thermo Fisher Scientific, Merelbeke, Belgium), Fungizone (1 μg/mL; Thermo Fisher Scientific, Merelbeke, Belgium) and Gentamycin (20 μg/mL; Thermo Fisher Scientific, Merelbeke, Belgium). An 80–90% confluent cell culture flask (175 cm^2^) was inoculated with 200 μL LSDV (10^5^ TCID_50_/mL) in 20 mL growth medium (DMEM + 2% FCS containing Fungizone 1 μg/mL and Gentamycin 20 μg/mL) and incubated for 4 days at 37 °C in the presence of 5% CO_2_. Following a freeze/thaw cycle, the virus/cell suspension was centrifuged (3000 rpm for 10 min), the supernatant was collected, aliquoted and stored in liquid nitrogen.

The samples for the current study were taken from two groups of five animals serving as infection controls in a vaccination experiment analogous to the experiments described by Haegeman et al. (2021) [21]. In brief, the animals were approximately 6-month-old Holstein bulls from Belgium, being free of LSD and where LSD vaccination is not practiced. The animals were tested to ensure they were free of LSD, BVD, Infectious Bovine Rhinotracheitis (IBR) and bluetongue virus and antibodies prior to purchase. The animals were inoculated with 6 mL LSDV suspension (LSD/OA3-Ts.MORAN; titer 10^7.5–8^ TCID_50_/mL), which was delivered intravenously (5 mL, *Vena jugularis*) and intradermally (1 mL). The intradermal injection was performed on two locations on both sides of the neck (250 μL per site) [21]. Following the infection, the animals were monitored for 21 days and sampled at regular intervals as described below.

### 2.2. Clinical Observations, Sample Collection

The animals were clinically evaluated daily throughout the entire trial. Body temperatures were scored daily. The onset of prolonged fever was defined as a body temperature of 39.5 °C or more for 2 or more consecutive days. Other daily observations were scored as described in Table 1 and used to calculate a cumulative clinical score. An animal with a subclinical LSD infection was defined as having no cutaneous lesions in which LSDV could be detected [5,22].

The samples for laboratory evaluation were collected on 5 different sampling days according to the schedule in Figure 1. EDTA blood and three different skin biopsies of normal skin (no lesions or nodules) were collected for the real-time PCR analysis. A first skin biopsy was taken in the neck at least 25 cm from the inoculation places, and a second one on the dorsal back area of the animal, both using Biopsy Punches (SMI, Sankt Vith, Belgium) with a diameter of 6 mm, following disinfection of the biopsy area and local anesthesia using procaine 2%. Subsequently, the biopsy lesions were closed using staples. The third sample was collected in the ear using ear notch punches (Allflex, Bad Bentheim, Germany) [19].

### 2.3. Molecular Analysis

DNA was extracted from EDTA blood using the Nucleo Spin Blood kit, and from biopsies and ear notches using the Nucleo Spin Tissue kit, both according to the manufacturer’s instructions (Filter Service, Eupen, Belgium).

LSDV DNA was detected using the panCapripox real-time PCR (RT-PCR) panel, consisting of three RT-PCRs (D5R, E3L and J6R) each with an internal (IC) and external control (EC), as described by Haegeman et al. (2013) [23]. Samples were determined strongly positive at a Cp-value <= 35.00, and moderately positive at a Cp-value > 35.00 and <=40.00. Samples with Cp values > 40.00 and <45.01 were considered doubtful, while samples with Cp values equal or above 45.01 were considered negative. Following an initial screening with the D5R real-time PCR, samples were analyzed using the E3L and J6R RT-PCRs if (i) the result of D5R was doubtful, or (ii) at conversion points (the PCR status of the animal changes from negative to positive or vice versa). A sample was considered low positive if it scored doubtful on a minimum of 2 of the 3 RT-PCRs.

### 2.4. Serological Analysis

The sera samples were analyzed using the immunoperoxidase monolayer assay (IPMA) as described in Haegeman et al. (2020) [24].

### 2.5. Statistical Analysis

Statistical analysis was performed using Prism, version 9.0.0; Software for Statistical Analysis; GraphPad Software, San Diego, CA, USA, 2020. Because of the early humane endpoint reached by one animal in this study (R02), instead of using repeated-measures ANOVA to compare the data of the two separate experiments, a mixed-effects analysis, followed by Šídák’s multiple comparisons test, was used to deal with the missing values for R02 [25].

### 2.6. Ethical and Biosafety Approval Code

This study was authorized and supervised by the Ethical and Biosafety Committee of Sciensano, Brussels, Belgium, under approval code 20150605-01 (Approved 23 March 2017) and Dossier_BV_2016_33_Durimm_08122016 (Approved 5 January 2017), respectively. The approved euthanasia protocol was according to the European Union and Belgian regulations on animal welfare in experimentation, briefly: after sedation, the animal is restrained in a slaughter box to perform the anesthesia by captive bolt, followed by electrocution after wetting the head and chest. Exsanguination is performed after pulling up on the hind legs and severing the carotid artery.

## 3. Results

### 3.1. Clinical Observations

#### 3.1.1. Animal Survival

Four animals in the first experiment and all five animals in the second experiment reached the end of the trial. One animal (R02) in the first experiment had to be euthanized because it was immobile for 48 h on 15 days post infection (dpi).

#### 3.1.2. Body Temperature

The body temperatures of all animals in both experiments are presented in Figure 2. No significant difference in body temperatures on any of the follow-up days was observed between the two animal experiments using the mixed-effects analysis followed by Šídák’s multiple comparisons test. The onset of prolonged fever was on 7 dpi and lasted for a minimum of 6 days in all animals, except for one animal in experiment 1 (R04) and two animals in experiment 2 (R07 and R08), with only 2 days of fever. The median maximum body temperature was 40.8 °C (40.4–41.1 °C) (min-max range).

#### 3.1.3. Clinical Scoring

Cumulative clinical scores, based on the scoring system described in Section 2.2, are presented in Figure 3 for all animals in both experiments. No significant difference in the cumulative clinical scores on any of the follow-up days was observed between the two animal experiments. In the clinical diseased animals, the nodules appeared on 6–7 dpi, followed on 7–8 dpi by the slight to severe loss of appetite for more than 4 consecutive days, and mostly mild but prolonged nasal discharge. The nodules became generalized by 10–12 dpi. One animal in experiment 1 (R04) showed only a slight decrease in appetite and a very mild nasal discharge on 1–2 days but no other clinical signs, including no nodule formation. Two animals in experiment 2 (R07, R08) also remained without nodules, and their clinical signs were comparable to R04.

#### 3.1.4. Clinical Versus Subclinical Infection

As no significant difference regarding the presence of clinical signs was found between the two experiments, all animals were analyzed for clinical signs together. Seven animals showed typical clinical signs of LSD, while three out of ten animals remained subclinical. The subclinical animals did not develop nodules. Instead, they only showed a fever for a few days, and showed very few and mild symptoms not specific for LSD. Figure 4 shows the median body temperature and median cumulative clinical scores of the clinical versus subclinical animals. The body temperature curve is similar for the clinical and subclinical animals for the first 5 days pi, reaching a peak by 8 dpi, followed by a prolonged fever for the clinical animals in contrast to a marked decrease from 9 dpi onward for the subclinical animals. After the fever peak, the mixed-effects analysis, followed by Šídák’s multiple comparisons test, showed a significant difference in body temperatures between the clinically and subclinically infected animals from 10 dpi (*p* = 0.033). The difference in cumulative clinical scores between the clinical and subclinical animals was significantly different from 8 dpi onward (median *p* = 0.003 (<0.0001–0.012)).

### 3.2. Molecular Analysis

#### 3.2.1. EDTA Blood Samples

No significant difference was observed in the Cp values of the blood samples on any of the sample days between the two animal experiments. The RT-PCR Cp values for all animals in both experiments are presented in Figure 5. The median and range of the maximum Cp values of the blood samples from the clinically infected animals are given in Table 2, as well as the maximum Cp values for the subclinically infected animals. All animals with clinical LSD reached peak blood levels of viral DNA between 12 and 16 dpi. In the subclinically infected animals, detectable levels of viral DNA were found at 5–8 dpi in the blood of R04 and R07 and at 8 dpi for R08 (Table 2).

#### 3.2.2. Skin Biopsies and Ear Notches

No significant difference in the Cp values on any of the sample days was observed between the two animal experiments. The RT-PCR Cp values of all animals in both experiments are presented in Figure 6a for the skin biopsies taken from the neck, in Figure 6b for the dorsal back area biopsies, and in Figure 6c for the ear notches. The median and range of the maximum Cp values from the clinically infected animals are given in Table 2, as well as the maximum Cp values for the subclinically infected animals.

Viral DNA was found in all animals in the skin biopsies and ear notches on at least two occasions. All animals showed the highest levels of viral DNA in the neck biopsies taken on 21 dpi. Animal R02 showed the highest level in the last biopsy taken before the animal reached the humane endpoint (12 dpi). LSD viral DNA was found in the neck skin biopsies of the subclinically infected animals R07 and R08 from 8 dpi on, and in all subclinical animals from 16 dpi until the end of the trial. The highest levels of viral DNA in the back skin samples of one clinically infected animal was found at 12 dpi, while in the other clinical and in the subclinical animals, the highest levels were detected at 16 or 21 dpi. The highest levels of viral DNA in ear notches of four out of seven clinically infected animals was found at 12 dpi, and in the other clinical animals, it was at 21 dpi. As shown in Figure 6c and comparable to both kinds of skin biopsies, the levels of viral DNA in the ear notches taken from the clinical and subclinically infected animals evolved from low at 8–16 dpi to strong positive Cp values in the samples taken at 21 dpi.

#### 3.2.3. Subclinical Versus Clinical Infection

The earliest samples in which viral DNA could be detected were the blood samples, with PCR positives in 6 out of 10 animals on 5 dpi. When analyzing the blood of clinically infected animals only, viremia was found in all seven animals within 8 days of experimental infection, and the samples remained PCR-positive until the end of the trial. Subclinically infected animals had detectable levels of viral DNA in the blood between 5–8 dpi only during a short period. The skin biopsies (neck and back area) and ear notches were PCR-positive for all clinical and subclinically infected animals from 16 dpi and remained positive.

### 3.3. Serological Analysis

The IPMA scoring for all animals in both experiments is presented in Appendix A. On 12 dpi, all animals—clinical or subclinical—were weakly to strongly positive. By the end of the trial, all clinical animals and one subclinical animal became strongly positive, while the two other subclinical animals were only weakly positive to positive (in dilution 1:50).

## 4. Discussion

LSD typically induces characteristic skin nodules and lesions on mucosal surfaces. The mechanical transmission of LSDV from clinically infected animals by several blood feeding insects has been demonstrated [5,6,26,27,28]. However, LSDV infection may range from clinical disease (severe and generalized) to subclinical (asymptomatic). Even when animals are experimentally infected, subclinical infection can occur in a significant proportion of the animals [6]. A recent quantifying and modeling study estimated that the role of subclinical cattle in LSDV transmission is minimal relative to clinical cattle [6], but the real importance of subclinical infection in transmission of LSDV in vivo should still be determined. As Haegeman et al. (2021) [21] found 67% of unaffected skin biopsies from subclinically infected animals to be positive for LSD viral DNA by RT-PCR, and because ear notch testing has shown its usefulness in BVD surveillance for many years [18], the current study aimed at investigating the applicability of testing normal skin biopsies and ear notches to detect the presence of LSDV in clinically and subclinically infected animals.

Until now, countries have not succeeded in eradicating LSD without vaccination [29], even when complete stamping out was applied in affected herds. Countries or regions where only clinical diseased animals were removed from a herd during an LSD epidemic were confronted with reappearance of LSD. The diagnosis of LSD is mainly based on clinical surveillance, complemented by laboratory confirmation [7] on the biopsies of nodules or blood samples. The detection of subclinical infected animals is more complicated, as nodules are absent, and blood samples are frequently negative because the viremia is short and/or intermittent [6]. In addition, in this study, it was shown that, in contrast to clinical animals, the sensitivity of the RT-PCR on blood samples for the detection of subclinically infected animals is low, confirming the findings of previous studies [5,6,21]. However, the capability of detecting the presence of subclinically infected animals in affected herds could be of importance in some countries to avoid whole herd slaughtering or to avoid the reoccurrence of LSD when only clinical diseased animals are stamped out [14]. The possibility to detect subclinical animals could be also of importance to substantiate the freedom of disease and regain trade possibilities [30]. Therefore, easily applicable alternatives to blood samples with a higher sensitivity for detecting subclinical diseased animals are needed.

In the current study, the seven animals in the two experiments that became clinically infected showed all the characteristic clinical signs of LSD, including generalized nodules. The remaining three animals showed the typical fever peak at 7–8 days post experimental infection but no nodules or lesions. The mild nasal discharge and reduced appetite they showed would most likely go unnoticed in the field. In this study, the antibodies found in the subclinical animals, together with the LSD viral DNA present in blood and in the skin biopsies and ear notches, provided proof of a successful experimental infection and, therefore, of the true subclinical status of these animals. The total clinical score of the LSD clinically infected animals surpassed 3 and rose to 6–9, while the score for subclinical animals did not surpass 2, which is a clinical evolution comparable to the patterns described previously [21]. It is known for LSD that the number of clinical animals can vary between 30–70% [29]. Less is known about the prevalence of subclinical animals. Recently, more data have become available, with studies classifying between 50% and 63% of the LSDV infected animals as subclinical [5,6,21,31]. In the current study, the prevalence of subclinical animals was 30%, which is possible given the high variability of clinical LSD animals [29].

In search of alternative samples for LSD testing, this study compared the outcome of the RT-PCRs on blood samples, biopsies of normal skin from the neck and the dorsal back area, and ear notches originating from clinical and subclinical infected animals. Since the animals were experimentally infected by injection of the virus suspension into the neck region, the viral DNA detected in the neck could be residual from the initial infection. For this reason, the biopsies were taken at a minimum distance of 25 cm from the injection spot, and supplementary biopsies were also taken from the dorsal back of the animals. Detectable levels of viral DNA were also present in the dorsal back samples of all animals on at least 2 consecutive sampling days, thus confirming the results of the neck biopsies. It is obvious that biopsies from the nodules and lesion crusts are still the samples of choice for the early detection of LSDV and/or the confirmation of a clinical suspicion. Blood samples could also be suitable, as LSD viral DNA was detected from 5 dpi on, and all clinical animals were detectable by 8 dpi. For early detection, biopsies of normal skin and ear notches are less fit for purpose, as LSDV DNA is detectable in most animals only several days later than in blood. On the other hand, blood samples are less advisable for the detection of subclinical animals, as the viremia can be easily missed in these animals because of the intermittent detectability, as shown previously [6]. In contrast, the ear notch samples and biopsies of unaffected skin tested positive for LSD viral DNA in all subclinically infected animals by 16 dpi, and are therefore more suited samples for the detection of subclinically infected animals. Although the results of the three kinds of biopsies were comparable (Table 2), ear notches are preferable. From a logistic point of view, ear notch samples are easier to take, and farmers in several countries/regions are used to having ear notches taken in the framework of the animal disease control programs for BVD.

The detection rate of LSD viral DNA in the normal skin biopsies in our study is higher than the findings of Sanz-Bernardo et al. (2021) [6], who detected viral DNA in the biopsy samples of normal skin in one out of five subclinical calves at three timepoints. The skin biopsies in our study originated from two different areas on the body, and the Cp values, which varied from strong to moderate positive, were confirmed by the results from the ear notches. Our results are in line with previous findings by Haegeman et al. (2021) [21]. The results indicate that even in subclinically infected animals, the dissemination of LSDV to the skin occurs without the formation of characteristic nodules. This could have implications for LSD transmission. Although the vector transmission of LSDV from subclinical infected animals is considered unlikely [6], the results of this study warrant further in vivo transmission studies.

## 5. Conclusions

Based on the results of our current study, ear notches could be a suitable sample for the detection of LSD subclinically infected animals in the framework of (i) research studies to determine the prevalence of subclinical animals after the acute phase of an LSD outbreak; (ii) field studies to check whether LSDV is still circulating in a subclinical form 2–3 weeks after removing the clinically infected animals in an affected herd; (iii) surveys in a later phase to substantiate freedom of the cattle population from LSDV, which is very important to regain trade; (iv) studies to check whether the waiting period could be reduced to regain free status after a case of LSD, in a country or zone previously free from LSD and where a stamping-out policy is not applied or only partially applied; (v) further research of LSDV transmission from subclinically infected animals.

## Figures and Tables

**Figure 1 microorganisms-09-02171-f001:**
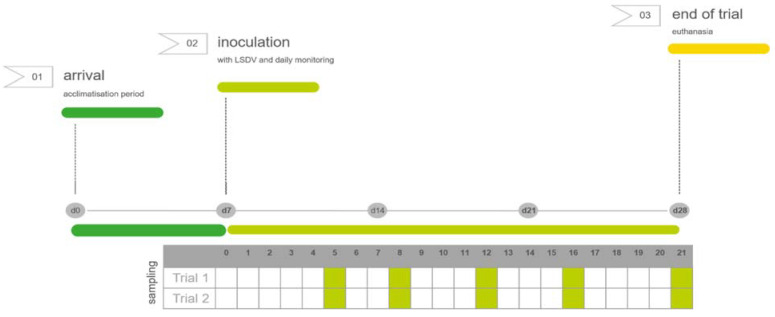
Experimental set-up: Following 7-day acclimatization (dark green), the animals were infected with LSD/OA3-Ts.MORAN and monitored daily for clinical signs (light green), until the end of the trial (yellow). Sampling was performed on the same selected days for both experiments.

**Figure 2 microorganisms-09-02171-f002:**
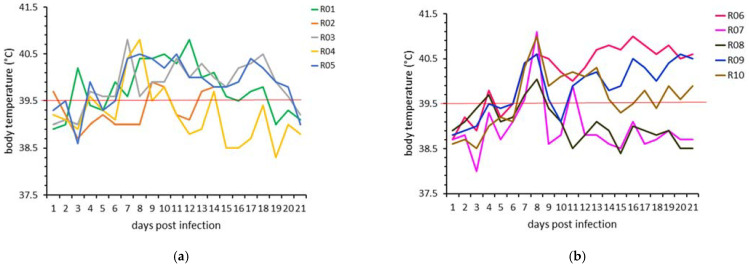
Daily body temperature of each animal in (**a**) experiment 1 and (**b**) experiment 2. Solid line: fever cut-off (39.5 °C) for prolonged fever.

**Figure 3 microorganisms-09-02171-f003:**
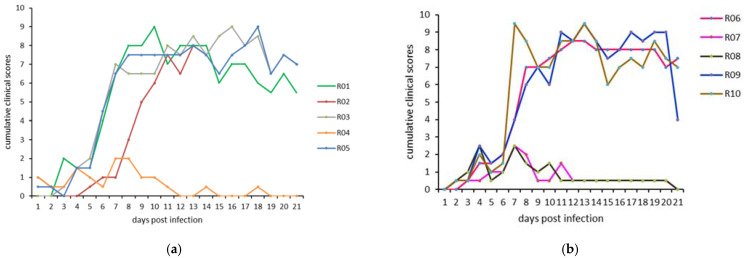
Daily cumulative clinical scores for each animal in (**a**) experiment 1 and (**b**) experiment 2.

**Figure 4 microorganisms-09-02171-f004:**
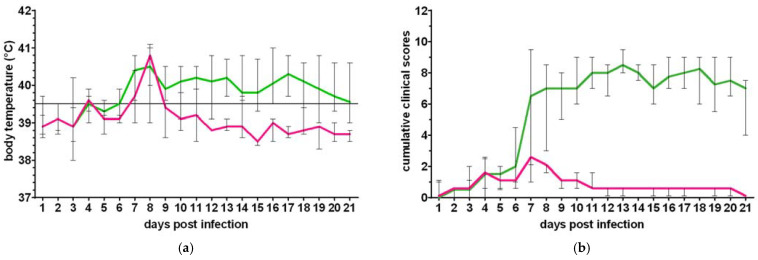
Median (min-max) body temperatures (**a**) and cumulative clinical scores (**b**) of clinically (green) versus subclinically (purple) infected animals.

**Figure 5 microorganisms-09-02171-f005:**
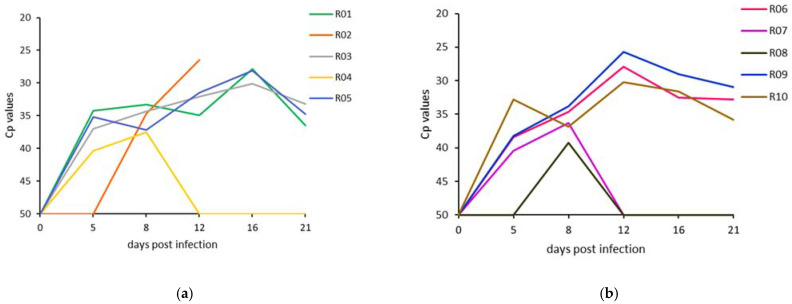
Real-time PCR Cp values for EDTA blood for each animal in (**a**) experiment 1 and (**b**) experiment 2 on consecutive sampling days.

**Figure 6 microorganisms-09-02171-f006:**
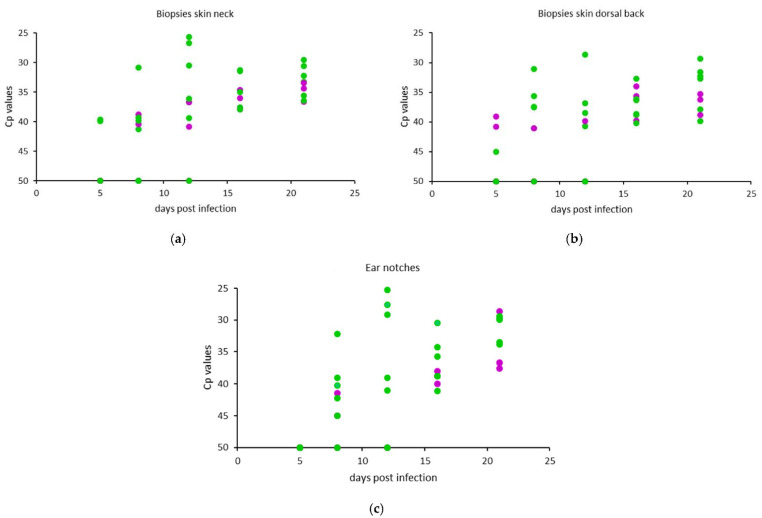
Cp values of (**a**) skin biopsies taken from the neck area, (**b**) skin biopsies taken from the dorsal back and (**c**) ear notches. The Cp values of the clinically infected animals are presented in green, and those of the subclinically infected animals are presented in purple.

**Table 1 microorganisms-09-02171-t001:** Clinical scoring system. The animals were monitored daily and scored using the scoring system in this table.

General Health Status		Food Intake		Nasal Discharge		Number of Noduli		Dissemination of Noduli	
Normal	0	Normal	0	Normal	0	No noduli	0	No Noduli	0
Mild Illness	1	Slightly Decreased	0.5	Mild	1	<10	1	Localized	1
Severe Illness	2	Decreased	1	Marked Mucous	2	>10	2	Generalized	2
		Does Not Eat	1.5	Purulent	3				

**Table 2 microorganisms-09-02171-t002:** The results of the real-time PCR for the blood, skin and ear notch samples taken in experiment 1 and 2, and given as the median and range of the maximum Cp values for the clinically infected animals and as the maximum Cp values for the subclinically infected animals.

Sample	Blood		Skin Biopsies of Normal Skin (No Lesions or Nodules).				Ear Notch	
Area	*Vena jugularis*		Neck Area		Back Area		Ear	
Experiment (Exp)	Exp1	Exp2	Exp1	Exp2	Exp1	Exp2	Exp1	Exp2
Clinical LSD, Cpmax at dpi (or Range)	(15–16) dpi	12 dpi	21 dpi	21 dpi	(12–21) dpi	21 dpi	(12–21) dpi	(12–21) dpi
Clinical LSD, Median Cpmax	27.96	27.88	33.50	30.62	33.82	32.24	27.06	29.41
Clinical LSD, Range Cpmax	(26.50–30.13)	(25.72–30.19)	(25.66–36.42)	(29.58–32.25)	(28.64–39.67)	(31.60–32.74)	(17.62–38.17)	(29.12–29.91)
Subclinical LSD, Cpmax at dpi	8 dpi	8 dpi	21 dpi	21 dpi	16 dpi	16 dpi	21 dpi	21 dpi
Subclinical LSD, Cpmax	R04: 37.53	R07: 36.28	R04: 36.61	R07: 33.47	R04: 39.77	R07: 35.60	R04: 28.61	R07: 37.54
		R08: 39.23		R08: 32.25		R08: 33.98		R08: 37.49

## Data Availability

The main data presented in this study are available within the study itself and other data may be made available through contact with the corresponding author.

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
