# Peer review of "Detection of Clinical and Subclinical Lumpy Skin Disease Using Ear Notch Testing and Skin Biopsies"

_microorganisms, 2021, doi:10.3390/microorganisms9102171_

Round 1
Reviewer 1 Report
The study is a nice contribution to the knowledge on LSDV. The experimental set –up is described in details and tests used show the dynamics of virus DNA. A solid justification of the proposed protocol is lacking.
Major concerns
No description of the euthanasia of animals
L300 “only clinical diseased animals are stamped out.” Can you please specify the countries? As far as I am aware, this practice has been discarded worldwide. Or delete if no reference.
L341. If a lactating cow is suspected, is ear notching it actually necessitated? I don’t agree that ear notching is an applicable alternative to blood samples. Should animals on quarantine be ear-notched twice or more to ensure LSDV freedom? Can you please provide a citation where adult animals are ear notched for testing way beyond birth. Ear notches are typically made within the first few days of birth. I could not find a rationale for taking ear notches to test for LSDV in already ear notched adult cows. I would not play down the importance of PCR in detecting subclinical animals because in case of suspicion more than one animal is sampled to rule out the infection be it clinical or subclinical.
Did the authors review all available literature with the relevant experimental designs and modes of inoculation? There are papers on the experimental evaluation of different LSDV strains.
L317 Do Moller et al [34] actually report subclinical infection? They report mild symptoms in three animals although skin samples from nodule lesions were positive.
L337 “more suited samples for the detection of subclinically infected animals.” I am afraid this claim does not work in the field. We have tried testing skin samples in naturally infected animals but nothing could beat blood and nasal samples. As for using PCR viral DNA may be missed in the blood, but nasal and conjunctival samples come very useful. Testing blood, nasal and oral samples is a proven way of detecting subclinical infection in the field. Never failed.
Moreover, it is arguable that ear notching caused less discomfort and stress than taking blood or swab.
L349 If the main mode of transmission is via vector bites and “the vector transmission of LSDV from subclinical infected animals is considered unlikely” why worry about subclinical infection and test ear notches??? In this regard the logic is as follows: clinically ill animals are tested by PCR, sublinically ill animals don’t play any significant role in transmission.
Conclusions should be moderated and limited to the reported findings without extrapolation and speculation. Without verifying the proposed approach in the field conditions, the true utility is impossible to objectively evaluate.
As I said earlier, a combination of blood, nasal and conjunctival samples detect subclinical infection in the field with 100% reliability. This cannot be pushed to the sideline in favor of ear notching.
Reviewer 2 Report
The study on the applicability of ear notch testing and skin biopsies in detection of LSD in clinical and sub-clinical animals was well carried out. The approach would help in detecting especially the sub-clinical animals increasing the sensitivity of clinical surveillance. The manuscript is well written presenting all the data and study aspects.
The legend of figure in supplementary information is incomplete without detail on N and + symbol.
